# Slowly Making Sense: A Review of the Two-Step Venom System within Slow (*Nycticebus* spp.) and Pygmy Lorises (*Xanthonycticebus* spp.)

**DOI:** 10.3390/toxins15090514

**Published:** 2023-08-22

**Authors:** Leah Lucy Joscelyne Fitzpatrick, Rodrigo Ligabue-Braun, K. Anne-Isola Nekaris

**Affiliations:** 1Nocturnal Primate Research Group, Department of Social Sciences, Oxford Brookes University, Oxford OX3 0BP, UK; 2Centre for Functional Genomics, Department of Health and Life Sciences, Oxford Brookes University, Oxford OX3 0BP, UK; 3Department of Pharmacosciences, Federal University of Health Sciences of Porto Alegre (UFCSPA), Avenida Sarmento Leite 245, Porto Alegre 90050-170, Brazil; rodrigolb@ufcspa.edu.br

**Keywords:** slow loris, venom, evolution of venom, venomous mammals, primates

## Abstract

Since the early 2000s, studies of the evolution of venom within animals have rapidly expanded, offering new revelations on the origins and development of venom within various species. The venomous mammals represent excellent opportunities to study venom evolution due to the varying functional usages, the unusual distribution of venom across unrelated mammals and the diverse variety of delivery systems. A group of mammals that excellently represents a combination of these traits are the slow (*Nycticebus* spp.) and pygmy lorises (*Xanthonycticebus* spp.) of south-east Asia, which possess the only confirmed two-step venom system. These taxa also present one of the most intriguing mixes of toxic symptoms (cytotoxicity and immunotoxicity) and functional usages (intraspecific competition and ectoparasitic defence) seen in extant animals. We still lack many pieces of the puzzle in understanding how this venom system works, why it evolved what is involved in the venom system and what triggers the toxic components to work. Here, we review available data building upon a decade of research on this topic, focusing especially on why and how this venom system may have evolved. We discuss that research now suggests that venom in slow lorises has a sophisticated set of multiple uses in both intraspecific competition and the potential to disrupt the immune system of targets; we suggest that an exudate diet reveals several toxic plants consumed by slow and pygmy lorises that could be sequestered into their venom and which may help heal venomous bite wounds; we provide the most up-to-date visual model of the brachial gland exudate secretion protein (BGEsp); and we discuss research on a complement component 1r (C1R) protein in saliva that may solve the mystery of what activates the toxicity of slow and pygmy loris venom. We conclude that the slow and pygmy lorises possess amongst the most complex venom system in extant animals, and while we have still a lot more to understand about their venom system, we are close to a breakthrough, particularly with current technological advances.

## 1. Introduction

When examining the convergence of certain traits across the Metazoa, few adaptations are as prevalent and persistent as venom [1,2,3,4,5]. Comprising a potent mixture of toxins, peptides, salts and other biochemical molecules, venom induces dramatic physiological effects like muscle paralysis or blood coagulation [2]. Although venom and poisons overlap as they can have similar toxic effects on their targets, animal poisons are delivered passively through secretions on the skin (Anura: *Dendrobates* spp.) [6] or within organs (Tetraodontiformes: *Takifugu rubripes*) [7], whereas the most recent definition of venom requires that the toxins must be “actively transferred to another organism through an injury by means of a specialised delivery system” [8]. The definition of venom is still contentious as we begin to learn even more about this intriguing trait, but this updated clarification has enabled novel taxa that would otherwise be dismissed due to using venom beyond predation or defence to fit the definition of venomous. 

The diversity of venom delivery systems and chemical composition in animals is staggering. Venom has evolved at least 101 times and currently has 14 identified functions such as predation, defence, offspring care and chemical communication [1,2]. There are over 250,000 species that use venom in some capacity, with a wide assortment of venom delivery mechanisms including spitting (Squamata: *Hemachatus haemachatus*) [9], spines (Scorpaeniformes: *Synanceia horrida*) [10] and ovipositors (Hymenoptera: *Pepsis grossa*) [11]. It is no surprise that with this huge array of venomous animal species, venom has been a consistently popular research topic in zoological, chemical and medical studies. Treatments to mitigate snakebite have dominated venom research: between 81,000 and 138,000 people die each year due to snakebite envenomation, resulting in the World Health Organization listing snakebite as a high-priority neglected tropical disease in June 2017 [12]. Toxins isolated from venom have been used to develop numerous novel products such as pharmaceuticals (i.e., Captopril, developed from jaraca snake venom (*Bothrops jaracaca*) [13]), pesticides (i.e., Spear-T by Vestaron, developed from Blue Mountains funnel web spider (*Hadronyche versuta*) [14]) and even cosmetics (i.e., SensAmone P5 by Mibelle Biochemistry developed from sebae anaemone (*Heteractis crispa*) [15]). 

Since the early 2000s, there has been a rapid development in another area of focus in venom research: the evolution of venom [1,2,3,4,5,8,9,10,16,17,18,19,20,21]. The large number of venomous species, delivery mechanisms and toxins suggest that venom appearing in a lineage is not necessarily restricted by environmental or physiological conditions (a notable exception being the complete absence of venom within Aves, see [21] for further details). This potential freedom for venom to evolve in vastly differing species offers a rare opportunity to observe evolutionary convergence (of both venom systems and composition [22]), real-time predator–prey interactions (due to the constant “arms race” species are under [23]) and investigate what epigenetic or post-translational processes lead to intraspecific differences in venom composition [24]. The understanding of how venom evolved in animals has been limited until recently due to the technology and equipment needed to examine toxins, genes and genomes to a detailed level. 

Prior to the 21st century, identification of toxins and other protein sequences in venom was limited to classic analytical methods (e.g., electrophoresis and gel filtration), focusing mostly on the molecular weight and amino acid sequences [25,26]. Understanding how potent these venoms were and what effects they had on humans was limited to lab experiments, injecting non-human models, followed by monitoring and observation of effects. This limited observation to a large range of median lethal dose (LD50) (depending on where venom is injected in an organism) and attempting to figure out if these LD50 are applicable to humans due to variable physiology compared to non-human models [27]. The creation of antivenom is vastly similar to how it was when it was first conceptualised in 1897—injecting a small amount of venom into a large animal, typically either a domestic horse (*Equus ferus caballus*) or sheep (*Ovis aries*), allowing the immune system to respond and then collecting the antibodies from the animal [28,29]. Now, access to affordable and rapidly advancing technologies are enabling these traditional practices to evolve and expand beyond their capabilities. 

Researchers are able to identify all isoforms of a toxin due to transcriptomics (which allows examination of RNA) and genomics (showing both genetic and epigenetic factors in venom evolution) [30,31]. Efficient and cheaper sequencing technology from Mass Spectrometry to HiSeq enables researchers to have the complete picture of a toxin for comparison or identification and even antivenom technological is shifting toward streamlining the process with organoids developed from venom glands, which would minimise the reliance on an individual animal’s need to regenerate venom [32]. These massive advancements, collectively known as the new field of “venomics”, have led to ground-breaking discoveries that have implications on the evolution of venomous animals. These include identifying major ontogenetic shifts in venom composition in cone snails (*Conus* spp.) [33]; discovering that venom within Araneae has evolved from silk glands rather than salivary glands [34]; and even the recognition of new venomous species in neglected groups like the mammals [35].

The mammals represent a controversial group within venom research as the scientific community still disagrees on which species are venomous. There have been numerous issues cited for this uncertainty, stemming from the difficulty in obtaining adequate samples, the protected status of some venomous mammals, alongside disputes on whether venomous mammals exist at all [36,37]. Understanding the venom systems of mammals may have untapped potential for deciphering the evolution of venom unlike any extant group [17,36]. Compared with most other venomous groups (i.e., snakes or cone snails), where venom has evolved once or twice in a recent common ancestor, venomous mammals have an unusual distribution where venom is represented within four completely unrelated groups within the Order [38]. This trait distribution offers an unprecedented opportunity to look at convergent evolution to examine homologous genes, regulatory networks and gene precursors to toxin genes [17,39]. 

It is generally accepted that there are fewer than 20 confirmed venomous mammals, which have different venom delivery systems and functions [1,19,36,37,38,40,41,42]. The best studied is the platypus (Monotremata: *Ornithorhynchus anatinus*), in which only the male is venomous, using his venom for intraspecific competition against other males to gain access to females in the breeding season [43,44]. Venom is more widely distributed amongst Eulipotyphyla, including solenodons (*Solenodon paradoxus* and *Atopogale cubana*) [18,43] and some shrews (*Blarina brevicaudua*, *Blarinella quadraticauda*, *Neomys anomalus*, *N. fodiens* and *Sorex araneus*) [35,42,44,45,46], which deliver their venom secreted from salivary glands via modified grooves in their incisors to capture prey and keep it alive in “food larders” to eat later. Three genera of vampire bats (Desmodontinae: *Desmodus rotundus*, *Diphylla ecaudata* and *Diaemus youngi*) [47,48] secrete their venom from salivary glands, delivering it via cuts to the skin of their prey, with venom facilitating and supporting blood feeding. The final taxa are among Primates and include the slow (*Nycticebus* spp.) and pygmy lorises (*Xanthonycticebus* spp.), whose venom systems form the basis of this review [49]. 

The slow and pygmy lorises (Primates, Strepsirrhini, Lorisinae) (hereon referred to collectively as slow lorises) are nocturnal primates with at least ten species in two genera distributed across southeast Asia (*Nycticebus bengalensis* Lacépède 1800, *N. bancanus* Lyon, 1906, *N. borneanus* Lyon, 1906, *N. kayan* Munds, Nekaris and Ford 2013, *N. menagensis* Lydekker 1893, *N. coucang* Boddaert 1785, *N. hilleri* Stone and Rehn, 1902, *N. javanicus* Geoffroy Saint-Hilaire 1812, *Xanthonyticebus pygameus* Bonhote, 1907 and *X. intermedius* Dao Van Tien, 1960) [50,51,52,53]. Several morphological traits characterise slow lorises, partly linked to their dedicated tree dwelling nature and specialised diet of exudates [54]. Lacking a tail, they use their strong limbs to hang on to and suspend themselves between branches, often for long periods of time, aided by a special network of blood vessels in their ankles and wrists [55]. They have modified anterior dentition that form a strong stout “toothcomb” allowing them to gouge exudates from trees by forming hundreds of holes (Figure 1) [56]. A long gut passage time allows for digestion of this highly fibrous and often toxic material, though they also supplement their diet with nectar, invertebrates, vertebrates and miscellaneous plant matter [57]. Their coat colour and face mask vary from white to reddish to dark brown depending on species but age and seasonality can change the intensity [58,59]. All species have a dorsal stripe, which in at least three species changes seasonally [60]. Other notable slow loris adaptations include the ability to undergo torpor, ultrasonic vocalisations and a unique specialised venom system [49,61,62].

The venom system of the slow loris consists of two components, the brachial gland exudate (BGE), generated by a modified sebaceous gland in the upper arm, and saliva, delivered by a bite via their toothcomb, which has grooves allowing the liquid to flow in a capillary action away from the mouth [57,63,64]. When a slow loris is threatened, it may exhibit a “venom pose”, raising its arms above its head and clasping them together, allowing the BGE to be taken quickly into the mouth. The venom of slow lorises acts as a two-step system, deployed by combining the BGE with their saliva. This mixture can be toxic to other mammals (including humans), inducing mild to severe symptoms, fatal to both conspecifics and invertebrates [65,66,67]. This two-stage external venom system is the only documented example known—venom systems are typically an internal gland linked to a fang, stinger or spur that requires no additional external activation to become toxic [2]. Due to the complexity of this one-of-a-kind venom system, the overlapping functions and the lack of available samples to characterise data, it has been difficult to discuss and review the slow loris venom system [64]. As a result, while we provide as much evidence as possible for many of the points disscussed within this review paper, we clarify that there are still aspects of the slow loris venom system that need to be confirmed with further observations or data. It is also worth mentioning that due to how unique the slow loris venom system is, it not necessarily an ideal comparison to other species that have an explicit venom system (i.e., snakes or spiders). Despite this, concise behavioural observations, new protein modelling and research published on other venomous animals enables us to review and discuss the current available knowledge on this specialist venom system. This review is particularly relevant as the last slow loris specific venom review was published a decade ago [49], and we aim to encourage further work on this exciting intersection of evolution and venom research.

Here, we review the research, including novel data collected since 2012, to address five topics about the slow loris venom system:The morphology and the two components (BGE and saliva) of the slow loris venom system.Functional usages of the slow loris venom system.The toxicity and mechanisms behind the slow loris venom system.The possible selection pressures that have caused slow lorises to evolve venom.The proposal of a research agenda for future research into the slow loris venom system to aid in providing focused and structured research.

## 2. Morphology and Components of the Slow Loris Venom System

### 2.1. Brachial Gland

The brachial gland used in the two-step venom system is a modified sebaceous gland located on the ventral side of the arm on a slow loris, within easy reach for the animal to lick [63,67,68]. Amongst the Lorisiformes, slow (*Nycticebus* spp.), pygmy (*Xanthonycticebus* spp.) and slender (*Loris* spp.) lorises are distinct in the presence of brachial glands, as opposed to pottos and galagos, which exhibit anogenital glands [69]. Early work by Montagna on slow loris brachial glands identified that the glands are characterised by high levels of phosphorylase activity and contain large amounts of succinic dehydrogenase, monoamine oxidase, and cytochrome oxidase [70]. Montagna also noted limited alpha naphthol esterase, some AS esterase, much Tween esterase, little acid phosphatase but much alkaline phosphatase. Even a human can detect the strong smell of the yellowish fluid exuding from these glands, which Montagna, who studied the skin and glands across the Order Primates, called richly innervated and “gigantic” in comparison to those in other strepsirrhines [71]. 

The underside of the region contains a mass of glands, 2 to 3 mm thick, with the upper side of the brachial gland of slow lorises covered with sparse fur [71]. Lorises under extreme stress or those kept in non-optimal captive conditions may exude excess oil leading to the gland becoming bald [59]. The secretion itself is liquid but can crystalise on the fur and skin; animals in hand, even in potentially stressful circumstances such as being captured for fitting with a radio collar or caught by a zookeeper for a health check, do not always produce BGE, suggesting the ability to meter the liquid [59]. Amongst the variety of olfactory communication systems used by primates, glands such as the brachial gland are most commonly used in chemical communication. Due to a moist rhinarium, the use of olfactory communication is pronounced among strepsirrhines [68]. Scent marking within the environment or on other conspecifics can signal rank, sexual availability, kinship and territory [72,73,74,75]. Scent glands can be found across the genitals, cranium and limbs of many primate species [68]; thus, an understanding of how and why this gland is used within the context of the venom system is intriguing (Figure 2).

### 2.2. Components of the Brachial Gland 

Mass spectrometry of the BGE from both a captive pygmy loris (*Xanthonycticebus* spp.) and captive Bengal slow loris (*N. bengalensis*) identified a wide variety of compounds found within it [76]. Approximately 212 compounds were found in the pygmy loris and 86 compounds within the Bengal slow loris, with 33 (13 named) of these compounds shared by both species (Table 1). Examining the chemical composition of the 13 named molecules found in both species identified that all compounds range from soluble to moderately soluble, which is consistent with data on other mammal scent glands. Four of the compounds identified fall into a group of high (Class I–III) toxicity risk: acetic acid, benzaldehyde, m-cresol and phenol, with benzaldehyde having potential carcinogenicity. A number of the named compounds (including the top four toxic entries in the table) are also found within other non-venomous mammals, suggesting that the brachial gland still plays an important role in chemical communication for slow lorises. It is also highly likely these molecules are non-toxic in the minimal amount they are secreted in the brachial gland.

Both loris BGE profiles [76] were noted for having minimal levels of phospholipids and other fatty acids (such as squalene or hexanoic acid) that appear in the majority of other olfactory glands in primates. This is indicative in the appearance of the slow loris BGE (see Figure 2) that can crystalise onto the fur and skin of the loris [77].

**Table 1 toxins-15-00514-t001:** Compounds of the brachial gland exudate identified by mass spectrometry; examples of molecules found in other mammal glands are a non-exhaustive list.

Molecule	MW	Consensus Log P	Ali Class	Predicted LD50 (mg/kg)	Predicted Tox Class Toxicity	Potential Toxicity Endpoint	Examples Found in Other Mammal Olfactory Glands
acetic acid	6005	−9	Very soluble	333	1	None	Yes, short-beaked echidna (*Tachyglossus aculeatus setosus*) [78]
benzaldehyde	10,612	157	Very soluble	28	2	Carcinogenicity	Yes, ring-tailed lemur (*Lemur catta*), Iberian red deer (*Cervus elaphus hispanicus*) and tamarins (*Saguinus imperator* and *Leontocebus weddelli*) [72,74,79]
m-cresol	10,814	177	Soluble	242	3	None	Yes, African bush elephant (*Loxodonta africana*) and Iberian red deer (*Cervus elaphus hispanicus*) [79,80]
phenol	9411	141	Very soluble	270	3	None	Yes, ring-tailed lemur (*Lemur catta*), African bush elephant (*Loxodonta africana*), Iberian red deer (*Cervus elaphus hispanicus*) and short-beaked echidna (*Tachyglossus aculeatus setosus*) [72,78,79,80]
1-heptanol	11,620	208	Soluble	1000	4	None	Yes, domestic sheep (Ovis aries) [81]
2-heptanol	11,620	199	Soluble	1000	4	None	N/A
anti-2-methyl-butyraldehyde-oxime	10,115	126	Very soluble	2000	4	Mutagenicity, Carcinogenicity	N/A
syn-2-methyl-butyraldehyde-oxime	10,115	123	Very soluble	2000	4	Mutagenicity, Carcinogenicity	N/A
n-butane	5812	205	Soluble	2000	4	none	N/A
dodecyl-aldehyde	18,432	394	Moderately soluble	5000	5	Estrogen Receptor Alpha (ER) activation	N/A
3-octanone	12,821	233	Soluble	5000	5	Estrogen Receptor Alpha (ER) activation	N/A
6-methyl-hepten-2-one	12,620	207	Very soluble	2400	5	none	N/A
2-methyl-butyraldehyde	8613	120	Very soluble	2490	5	Carcinogenicity	N/A

MS results are taken from [76]. Pharmacokinetic properties were estimated with SwissADME [82], toxicity was predicted with ProTox II [83].

### 2.3. BGE Secretion Protein (BGEsp) Only 

As of 2023, only one component of the brachial gland exudate has been identified in detail from a single species of slow loris (*N. javanicus*)—the BGE secretion protein (BGEsp, Figure 3) [64,84,85]. The BGEsp is a 17.6 kDa homodimer formed of two heterodimer chains that are linked together by three disulfide bridges each. This means the BGEsp has two parts that are made of two polypeptide chains that have a different structure (Chain A1 and A2, 7.8 kDa, vs. Chain B1 and B2, 9.8 kDa), but these two parts are almost mirror images of each other when arranged in the tetrameric form (dimer of dimers). The two chains that make up the BGEsp halves are called Chain A and Chain B (see Figure 3). The BGEsp is part of the secretoglobin family, a group of proteins exclusively found in mammals. It was first visualised in [64]; here, we present a model with all four chains clearly defined. 

Our understanding of secretoglobins is limited; their name is derived from the illusive nature of these proteins [88]. What is understood is they have a possible variety of functions, including transporting ligands and within humans, they have a potential role in preventing diseases within the lungs [88,89]. BGEsp is not the only secretoglobin identified in mammal venom: non-toxic secretoglobins have also been identified within the venom of the common vampire bat (*Desmodus rotundus*), and a similar product is found in short-tailed shrew venom (*Blarina brevicauda*) [42,47]. Secretoglobins have a prominent link between olfactory communication and allergies—the majority of secretoglobins described within the mammals belong to the “Major Allergen 1”family. The reason for secretoglobins resulting in so many allergies is proposed to be because they mimic lipocalins, which are an important part of the immune system [90]. Where this becomes even more striking is the structural and functional homology of BGEsp compared to Fel-d-1, the most common secretory protein produced by domestic cats (*Felis catus*) and the reason for most cat allergies within humans [64,84,85,91]. Indeed, we put Chain A and Chain B from [64] through BLASTP [92] to determine which sequences are homologous (Appendix A). Both Chain A and Chain B had greater similarity to Felidae major allergen proteins (including Fel-d-1) than primate major allergen polypeptide chains. Chain A specifically displayed greater homology to all Felidae, while Chain B displayed high homology levels only with Fel-d-1 from domestic cats. This convergent evolution is highly unusual between the slow lorises and Felidae, as it brings into question a tentative link between venom and allergies [85]. The purpose of Fel-d-1 within cats is still not currently understood, but it is considered important in chemical communication between cats [93]. 

### 2.4. Saliva

The two-step venom system relies on BGE being combined with saliva. Alterman [63], for example, found that loris venom was only significantly deadly to mice when the two were combined. The injection mechanism of the saliva, with or without BGE, is the toothcomb, which, as mentioned above, has a capillary action for liquid to flow out of the mouth (Figure 4) [49]. Similarly, it was found that saliva was necessary for the venom to kill ectoparasites [66]. Slow loris saliva seems to exhibit a level of toxicity in the absence of BGE. In a study of slow loris keepers, humans experienced numbness of extremities and blood in the urine upon simply touching the animal (or petting it), with other symptoms including lethargy and nausea. In keepers and field workers who were bitten when the animal did not have an obvious chance to lick the brachial gland, the main symptom reported is throbbing and pulsing pain, described as more extreme than a dog, cat or monkey bite ([65,94]; Nekaris pers. obs.). Such statements comply with the results of an unpublished study on Javan slow loris saliva [95]. Using electrophoresis and mass spectrometry, Aprilia [95] found that slow loris saliva differed substantially from human saliva, being dominated by a 25 kDa protein, which is also not present in the brachial gland exudate. This protein was identified as a complement component 1r (C1R) protein, which could contribute to causing excess inflammation, swelling, numbness and even anaphylactic shock. Another feature of the slow loris saliva system is the morphology of the ducts of the salivary glands, which contain giant granules delimited by a membrane, which are 1.5 times larger than the largest granules ever reported in animals of similar size [96]. The glands are also characterised by striated ducts containing kallikrein, in an abundance similar to that seen in solenodons and shrew saliva where it is considered a toxin [18,42,43]. This suite of traits suggest that the saliva is specialised for secretion rather than transport.

The reason why slow loris saliva is characterised by these traits may be linked to the unique diet of the species. Work on the saliva of other primates indicates that there is a significantly higher abundance of proteins (~3–4 mL/mg) in chimpanzee (*Pan troglodytes*), gorilla (*Gorilla gorilla gorilla*) and Rhesus macaque (*Macaca mulatta*) saliva compared to human saliva, which is also much more viscous [97]. These differences in viscosity are attributed to the rapid diversification of human diets over the last two million years when compared to predominately herbivorous or frugivorous diets of gorillas and chimpanzees that chew their food for longer to make it easier to digest [97,98]. Tree exudates like gum and phloem sap, the main foods of slow lorises, are even more metabolically intensive to digest than other plant matter. For example, in a study of three slow loris species, gut retention time ranged from 30 to 39 h, whereas transit time ranged from 24 to 26 h [99]. Thus, slow loris saliva may have even more proteins per ml/mg of saliva to promote digestion of tree exudates. Not only are exudates high in fibre, with the need to be digested slowly, but they also may be high in secondary compounds such as phenolics, which are known to heal skin in humans [100]. Exudates eaten by slow lorises are frequently used in human traditional medicines [101]. Slow lorises in rescue centres and zoos that lack exudates frequently die of bite wounds, whereas wild slow lorises, who have ready access to the phenolic compounds of exudates, are frequently able to heal [102,103]. Although an increase in saliva proteins may not necessarily indicate toxicity in saliva, the unique slow loris diet was one of many traits under intense selection pressure as the genus colonised Asia’s forests during the Miocene [104].

## 3. The Functional Usage of Venom in Slow Lorises

### 3.1. Ectoparasitic Defence 

Parasites are a serious threat to primate health as ecto- and endoparasites cause transmissible diseases, resulting in death, and can be difficult to eradicate [105]. Work by Solórzano-García and Pérez-Ponce de León found that out of the 22 genera representing 171 neotropical primate species, all but one genus (*Callibella*, now part of *Mico*) had multiple documented cases of parasites [106]. Slow lorises are no exception, with numerous endoparasite species found in wild and captive individuals such as *Blastocystis* spp. (*N. bengalensis* in Dehong Wildlife Rescue Center in Yunnan, China [107]), *Strongyloides* spp. (*N. menagensis* in Lower Kinabatangan Wildlife Sanctuary, Sabah, Malaysian Borneo [108]) and *Pterygodermatites nycticebi* (*N. coucang*, zoological garden in Hyogo Prefecture, Japan [109]). Yet, compared to other primates and confirmed in multiple wild studies across various species of slow loris, they are remarkably clear of ectoparasites such as ticks or fleas. In 2015, Rode-Margono et al. examined a population consisting of 21 wild *N. javanicus* (West Java, Indonesia) in an agroforest where wild boar, domestic goats, cats and dogs are prevalent. In 61 captures, they found an unknown ectoparasite in a single individual, which was causing rash and hair loss [110]. In the same population, the researchers measured 141 individuals (ranging from 1 to 20 times each) in 531 captures for health checks. They observed ticks only four times and fleas only once (Nekaris, unpublished data). This successful clean bill of health has been attributed to a functional usage of slow loris venom. Grow et al. examined the effects of wild *N. javanicus* and recently seized *N. coucang* venom on invertebrates (caterpillars (Lepidoptera), maggots (Diptera), fleas (Siphonaptera), spiders (Araneae) and ants (Hymentoptera) [66]). Of the 73 invertebrates tested, 76% were impaired immediately and death occurred 61% of the time after 1 h. There was no significant difference between the slow loris species venom potency suggesting that lorises kept their venom in captivity. They did find a significant difference between the number of spiders and maggots that died: 78% vs. 42%. The difference suggests that slow loris venom has a greater effect on spiders because it has been selected to target arachnid parasites such as ticks. Venom has had a number of toxins confirmed to have antiparasitic (and antimicrobial) qualities—the ponericin toxin, M-PONTX-Na1b, from the ant *Neoponera apicalis* prevents development of *Haemonchus contortus*, a nematode that causes anaemia and death in ruminants [111]. Despite this strong evidence to suggest that slow lorises use venom for ectoparasitic defence, we note that explicit confirmation of this functional use has not yet been determined as this would likely require a combination of laboratory studies, behavioural observations and comparisons with related species, a not insignificant task. 

### 3.2. Intraspecific Competition

One of the largest selection pressures is arguably intraspecific competition, whether it be for territory, resources or access to mates (see [112,113,114,115] for further discussion). Despite its importance as a major evolutionary driver, intraspecific competition as a selecting factor for venom is limited to only a handful of species [2]. Currently, intraspecific competition has only been observed in a few species within the phylum Cnidaria (i.e., Hydrozoa, Octocorallia and Hexacorallia) [116] and two other animals, both mammals: the male platypus and slow lorises [37]. 

For slow lorises, intraspecific competition seems to be the main driver of their venom system [67]. Indeed, female–female competition for territories characterised by defendable gum sources and male–male competition for females in a strictly uni-male uni-female social organisation, combined with slow locomotion limiting escape options, may be linked to selection for this trait. The first extensive documentation of such wounds came from Rasmussen, who described the large and severe wounds inflicted by slow lorises, leading to the death of two individuals at the Duke University Lemur Center [117]. He also noted that unlike other strepsirrhines, slow lorises clamp their jaws tightly and persistently usually on their opponent’s head or rump. Death from trauma was the third highest cause of mortality in slow lorises in a review of deaths in North American zoos from 1980–2010, including purulent abscesses, cellulitis and septicaemia from wounds [102]. In a confiscation of 77 *N. coucang* specimens from illegal wildlife trade, 33.1% died of wounds from bites [103]. These bites can develop into deep necrotic wounds to the head, anogenital region, and extremities, including stiff or bitten off fingers and toes, with the extreme result being death. 

This function of slow loris venom receives even more support from the long-term continuous study of a population of *N. javanicus* in Cipaganti, West Java, Indonesia. Like other slow lorises, Javan slow lorises live in uni-male, uni-female groups that are highly territorial [57]. Within 8 years (April 2012–June 2020), a total of 25 aggressive encounters between *N. javanicus* were documented, with one of these fights resulting in the death of an individual [67]. Six of these encounters were related to fighting off competing males from females (*n* = 4) or females rejecting mates (*n* = 2), with others linked to fighting with intruders. Of 48 *N. javanicus* captured and recorded on over 338 occasions in the same study [67], 33% of females and 57% of males had evidence of bite wounds present. This high level of conspecific trauma plays a key part in slow loris sociality. In 663 observations of play between April 2012 and December 2020 of the same population of *N. javanicus*, juveniles engaged in play fighting with adults on nearly 60% of occasions, allowing them an opportunity to test the limits on play and practice for future challenges [118]. In captures of *N. javanicus* between June 2011 and June 2018, slow lorises during capture who were measured as more “aggressive” (i.e., loud aggressive vocalisations, attempted biting and very active attempts to escape) were found to have significantly fewer fresh bite wounds. Males were typically observed to be more aggressive than females (although both engage in intraspecific competition) [59]. The latter study also revealed that change in the colour contrast of facemasks may be an aposematic signal of toxicity or aggressiveness, particularly during the time when a slow loris disperses and seeks a new stable territory. This sexual difference was also seen in other slow loris species; of 26 wild captured *N. coucang* specimens, 52% of males compared to 12% of females had fresh or healing bite wounds [119]. 

### 3.3. Intra- and Interspecific Chemical Communication Rather Than Defence

Predation and defence are the two most common functions for venom observed within species—with 65 venomous lineages having predation as their main or only functional usage for venom compared to 40 for defence [2]. Within slow lorises, predation on insects and invertebrates was tested as a potential function for the venom. In almost all cases, however, slow lorises simply bit the head off the proposed victim, with no evidence that venom aided in their powerful bite (Nekaris, unpublished data). If slow lorises can use their venom to kill other slow lorises, using it on predators still seems to be an avenue worth exploring.

Because of the strong evidence of venom for defence against predators, both Alterman and Nekaris et al. presented various carnivores with slow loris BGE mixed with saliva, leading to the repelling of cats (*Panthera pardis*, *Panthera tigris*, *Neofelis nebulosa*), sun bears (*Helarctos malayanus*), and civets (*Paradoxurus hemaphroditus*, *Arctictis binturong*) [49,63]. Nekaris et al. further presented the venom to Bornean orangutans (*Pongo pygmaeus*, *n* = 2), both of which ate the sample [49]. In a more structured study, changeable hawk eagles (*Nisaetus cirrhatus*, *n* = 10) and crested serpent eagles (*Spilornis cheela*, *n* = 5) were presented with venom samples alongside control samples [120]. Other than non-significant face rubbing, the birds displayed no sign of being repelled by the venom. Both birds of prey and orangutans are known but infrequent predators of slow lorises. There are only 15 documented predation events of Sumatran orangutan (*Pongo abelii*) in the Gunung Leuser National Park, northern Sumatra, Indonesia on *N. coucang* [121,122,123] and one record of predation by a Bornean orangutan (*P. pygmaeus*) at the Tuanan Orangutan Research Station, Central Kalimantan, Borneo, Indonesia on *N. borneanus* [123]. These attacks all occurred in the day, when slow lorises sleep curled up tightly in a ball. Because of the ease of detecting them and the very long-term nature of the orangutan studies, this low sample size over decades of research suggests that such predation is infrequent. The strong negative responses seen in predators to BGE and minimal records of predation have been attributed to the success of venom in slow lorises being used in defence. It cannot be forgotten, however, that slow lorises are highly cryptic, moving slowly with little disturbance to surrounding vegetation, communicating in pure ultrasound, and being less active in moonlight, all of which help them to actively avoid predators [57]. These traits alone seem more powerful than venom to avoid predation.

A final piece of evidence against the suggestion that the slow loris uses venom in defence is that the non-venomous slender lorises (*Loris* spp.) have a similar defensive response, wrapping their arms above their head and swaying like a snake whilst hissing. In 18 records of both slender loris species being present close to a predator, the typical response was to completely ignore the animal and resume activities, with only an encounter with a barn owl (*Tylo* sp.) having an active antipredator response (vocalisations and cover) [69]. 

We have two interpretations for this mixed bag of evidence, that instead of venom being used for defence:The strong negative reaction to BGE by olfactory predators, the smell described by humans in both bite cases and research, the similarity in non-venomous slender lorises, the lack of descriptions to suggest human bite cases have venom involved and that BGE has many components found in non-venomous species (see Table 1) do suggest that venom has no role in antipredator strategies. Similar to animals like mustelids, slow loris brachial glands could produce olfactory communication that wards off predators and may communicate fitness or presence to other slow lorises.Venom does play a role in antipredator defences, but it would be better described as intra- and interspecific communication. Referring to the similarity of BGEsp to Fel-d-1, post-transcript modification of Fel-d-1 has been observed in cats. With different types of glycosylation (a process in which different carbohydrate molecules can be bound to another molecule) of the protein depending not only on breed, age or sex but even under high levels of cortisol present within an individual cat [93]. This modification of Fel-d-1 not only alters the amount produced, but also produces different isoforms of the protein, with glycosylation being able to alter the ligand-binding cavity volume, providing additional layers of plasticity. These are assumed to have different signals, such as alerting other nearby individuals of potential stress or marking territory. It is possible that grooming near predators and licking of conspecifics in slow lorises may still be using venom in an additional functional way by using posttranslational modifications on BGEsp.

## 4. The Toxicity and Mechanisms behind the Slow Venom System 

The difficulty in understanding slow loris venom stems not just from what the venom system is composed of nor what the venom is used for, but also from how the venom works exactly. We still do not have an exact measurement of how much of either saliva or BGE is produced by slow lorises nor how much is required of either to become toxic. The toxicity of the slow lorises has been debated in the scientific literature as a result—even the research team that first reported the similarity between BGEsp and Fel-d-1 in 2003 put venom in quotation marks in the title, implying it was not perceived as true venom [84]. The definition of venom has changed over the years (see [8] for further discussion), with definitions often focusing on its relation to humans or fatality rather than functional usages, with slow lorises often left out of definitions as a result. Perception as to what makes a venomous system has also contributed to this prejudice in what we described as a venomous species, as the most popular examples of venomous animals (i.e., spiders, snakes and scorpions) typically feature a modified internal gland that secretes toxins and a very sharp delivery mechanism like a fang or sting [1,2,4]. Venom glands secrete toxins and non-toxins that work in tandem with the toxins acting as spreading agents, antimicrobials or activators, with venoms ranging from relatively simple compositions (i.e., short-tailed shrew, *B. brevicadua* [42]) all the way to enormous venom arsenals with thousands of identified molecules (i.e., funnel web spider, *Hadronyche infensa* [16]). Research on the effects of individual toxins and whole venoms can be performed by identifying symptoms on target organisms, the length of time symptoms have/how long it takes for death to occur and other measurable data such as the consistency of blood or secreted markers. 

The slow loris venom system in comparison is unlike anything seen in known venomous species: part of the venomous system is secreted from an external gland which also secretes potential unrelated chemicals, the other part of the venom system is secreted within an internal gland, and both parts must be combined together to achieve full toxic potential [49,64]. We are lacking critical pieces of the puzzle to decipher slow loris venom; we do not know exactly what parts of the venom system are responsible for observable symptoms of envenoming, we do not know how these parts are necessarily activated to become toxic, we are not sure of the full effects and symptoms that can be caused by the venom and there is even the issue of deciding whether we are seeing one or multiple venom systems working within slow lorises. This is particularly frustrating, as slow lorises are well represented by numerous high-quality genomes, even to a chromosome-level assembly. Genomes have provided numerous valuable insights into venom evolution (see [31] for examples of venomous snake genomes). However, to use these effectively would require transcriptomics and nucleotide sequences to ensure the correct identification of the genes involved in venom within the slow lorises. Yet it is these difficulties and incomparable morphology that makes the slow loris venom system tantalising as a model for understanding the evolution of venom. 

### 4.1. The Type of Toxicity

Broadly, the type of toxicity seen as a result of venom falls into four categories: neurotoxins (toxins that target the nervous system), cytotoxins (toxins that target cells, necrotoxins, which result in cell death can be seen as sub category of this), myotoxins (toxins that target muscles) and haemotoxins (toxins that target blood and the related circulatory system) [1]. Numerous other categories of toxins are also described in venom, but they are either subcategories of the main four or specifically identified only within a few venoms. Venoms are not limited to toxins found in one category; the best example of this is seen in snake venoms such as the fer-de-lance (*Bothrops asper*), which cause myotoxic, cytotoxic and neurotoxic symptoms [124]. 

In slow lorises, we see similar symptoms following envenomation in different species and media. Slow loris bites on each other result in slow healing wounds that develop necrosis, infection and cellulitis as commonly observed within wild *N. javanicus* (Figure 5) [59,67]. Captive populations of *Xanthonycticebus* spp. and *N. coucang* in North American zoos have also reported pulmonary edema in slow lorises bitten by conspecifics [102,125]. One infant also died from overgrooming, something that has also been reported anecdotally from several Southeast Asian rescue centres, where loris mothers not only over-lick their offspring producing severe necrosis, but adults can also over-lick themselves, which leads to death (Nekaris, pers.com).Unpublished work on *Xanthonycticebus* spp. saliva applied to human epidermal carcinoma cells (skin cancer) showed the death of mitochondrial functions using a MTT assay of the cells, even at very low (1/10,000 part saliva in water) dilutions [126]. Symptoms from slow loris bites on humans are well documented, with a broader variety of symptoms including lethargy, headaches, festering wounds, paraesthesia and infections [65,94,127,128,129,130,131]. Together, this evidence represents slow loris venom as predominately cytotoxic. 

Yet, within humans at least, there are other notable symptoms that cannot be explained as simply cytotoxic—a small number of cases have resulted in symptoms similar to those of anaphylactic shock, with swelling of the face and airways a particular diagnostic feature [65,128,131]. Due to the similarity BGEsp has to Fel-d-1, it was proposed that those with cat allergies were more likely to exhibit these more severe symptoms, and, indeed, in a bite case resulting in anaphylaxis in 2021 involving an unidentified species of slow loris, the patient did have a cat allergy [128]. This is not the case for every single slow loris bite that involves anaphylaxis, though; additionally, one person who had presented anaphylaxis after a slow loris bite could only continue to work with them if they used an antihistamine [65,131]. Tentatively, we propose that slow loris venom could be the first documented venom to have immunotoxin effects. There are a few cases of impaired blood coagulation and haematuria (blood in urine), which could also suggest haemotoxic venom, but this could also be due to potential anaphylaxis shock or the mechanical action of the bite [132].

### 4.2. Activation of the Toxicity

What triggers the activation of venom within the slow loris system? We propose two lines of thinking: that the exudativorous diet is responsible for toxicity or that a serine protease within the saliva activates glycosylation sites on the BGEsp.

#### 4.2.1. Diet and Its Influence on Venom

Tree exudates encompass organic materials obtained from trees such as sap, gums or resins—sap is the fluid that is found within the xylem or phloem of a plant that transports nutrients and water, gums are polysaccharides produced by some woody plants and resins are a mixture of organic compounds that act as a potential anti-predator defence [133,134]. An exudativorous diet is found in numerous primates, including common marmosets (*Callithrix jacchus*), forked-mark lemurs (*Phaner* spp.), needle-clawed galagos (*Euoticus elegantulus*) and all species of slow loris [54,56,135,136].

Slow lorises have been documented on multiple occasions eating exudates from plants that may possess cytotoxic chemicals or have other properties that make them dangerous/deadly for humans to touch or eat. We found 126 species that slow lorises have been found to consume in the wild, of which we identified 32 that had a documented detrimental effect on humans (cytotoxicity, hepatoxicity or other conditions) (Appendix A). We also identified 36 plants that slow lorises consume that may have medicinal properties. Some examples include *N. bengalensis* consuming the exudate of *Acacia concinna* (in Trishna Wildlife Sanctuary, northeast India) [137] and *Lannea grandis* (in Assam, India) [101], both of which have documented toxic effects, particularly cytotoxic skin effects and chemicals that can change the pH levels in the saliva [138,139]. 

A number of known chemicals found within plants can cause a drop in the pH of saliva—an example within humans being tannins found in tea, coffee or wine. The pH of saliva (in humans) is expected to be between 6.7 and 7.3 in a healthy individual, which is a neutral pH level [140]—consuming food and drink can alter the pH level in an individual (i.e., alcohol or carbonated soft drinks), but this is only a temporary alteration unless this it is a large part of the individual’s diet [141]. Within spider monkeys (*Ateles geoffroyi*), the presence of tannins induced a physiological response with increased protein concentration of the saliva and an increase in pH to become more alkaline (from 8 in control specimens to 8.77 in specimens eating tannins) [142]. Exudates consumed by lorises may increase the pH of the saliva to more alkaline levels, which could interact with properties of the BGE—when licked by a saliva with a changed pH, it may in turn react and become toxic. The chemicals in exudates may also interact with proteins within the saliva itself. As mentioned in Section 2.4, non-human primate saliva has a significantly higher proportion of proteins compared to that of humans [97]. The majority of acids, polysaccharides and esters that can be found within exudates are predominantly hydrophilic and dissolve readily in water, which in slow loris saliva could trigger a reaction that, when mixed with BGE, unlocks toxicity. 

There is also the possibility that slow lorises are sequestering toxins from exudates. Toxin sequestration is when an animal is able to weaponise toxins or chemicals from a prey source. The poison dart frogs (Dendrobatidae) diet of ants and other invertebrates contains chemicals (notably alkaloids) that contribute to the notorious toxicity of the frogs [143]. In vertebrate animals, an exogenous origin of toxins has only been documented within Amphibians and snakes [144]. The reason for this very limited distribution of a trait has been proposed to be due to the physiological toll it takes on organs, the specialised complex synchronicity between the different organs and the impact on molecule transportation [6,144]. It has also been suggested that the lower metabolism of reptiles and amphibians is a key factor in these species being able to sequester toxins [144]. 

A recent large-scale genomic study on the *N. bengalensis* and *Xanthonycticebus* spp. revealed a number of highly specialised genetic adaptations for both lower metabolic rates and coping with a toxic diet [145]. These included higher upregulation of proteins such as *ABC9* and *TRPC5*—which is also seen heavily upregulated in poison dart frogs that are fed a toxic diet compared to individuals on a non-toxic diet [6]. Slow lorises could therefore be sequestering toxins from the gum of trees that they eat. However, even in captivity, on diets with fewer to no exudates (and when included, they are non-toxic commercial exudates), slow lorises are capable of delivering toxic bites [65]. The genes upregulated are detoxification proteins that are also seen convergently in koalas (*Phascolarctos cinereus*) and sloths (Folivora). Neither of these animals are regarded as venomous, but both do consume plants that can be toxic (i.e., eucalyptus plants in koalas). Other genes upregulated in *N. bengalensis* and *Xanthonycticebus* spp. include *FM05* and *GDPD4*, which are involved in detoxification within the liver, but all four genes (*ABC9*, *TRPC5*, *FM05* and *GDPD4*) most popular gene ontology terms in their expanded gene families related to binding and not to transportation, implying that they are not specifically related to sequestering. The slow metabolism seen in lorises, similar to koalas and sloths, could be linked to a toxic diet releasing few calories and needing a longer time to process in order to remove toxins [99]. 

#### 4.2.2. Complement Component 1r Found within Slow Loris Saliva

As noted in Section 2.4, Aprilia [95] examined the saliva of *N. javanicus*, taken from multiple individuals in West Java, Indonesia. Using LC-MS/MS and SDS-PAGE, a total of 9051 proteins and 667 peptides were identified within the saliva, with one protein predominantly upregulated with the venom: Complement component 1r (C1r). C1r is involved in the complement immune response within humans—it is needed for a chain of reactions that starts phagocytosis, clears microbes or dead cells and attacks pathogen cell membranes [146,147]. The triggering of the complement immune system is also known to result in inflammation [146,147]. Additionally, C1r is a serine protease, able to cleave proteins via proteolysis [148]. The presence of C1r in saliva of *N. javanicus* is thrilling—this enzyme could itself be responsible for the anaphylactic symptoms due to the immune response it triggers. It could also hinder the targets immune system from properly working, which is why we see slow healing/infections in many slow loris and human bites. In addition, it may also be binding to the glycosylation sites on BGEsp which activates its toxicity or it may cleave the protein to result in toxicity. 

Unfortunately, a lot of the discussion around this discovery will remain speculative until it is repeated again and identification of if this protein is actually confirmed to be C1r expressed in the saliva or an isoform. We also note that within [95], a BLAST run on the proteins found in the saliva of *N. javanicus* was used to confirm the identity of the protein described as C1r—the majority of these hits came from birds (i.e., entry KFP92140, UniProt ID A0A091PKX8_APAVI, comes from *Apaloderma vittatum*, the bar-tailed trogon). There is also no evidence to support that C1r actively has an immunotoxic response, only that it exists within the saliva of *N.javanicus*. Finally, although C1r starts a cascade within the immune system, it cannot start without the presence of the specific individual antibodies activating the reaction, and there are proteins that inhibit the C1r present, too [149]. 

There does exist a possible venom-based solution to this issue of activation: a nontoxic component of cobra venom (notably described mostly from Asian species of cobra, *Naja naja* and *Naja kaouthia*), dubbed cobra venom factor (CVF), is homologous with component C3 and activates part of the immunological complement pathway in the immune system [150]. Laboratory research with isolated CVF demonstrated that it induces a dermal anaphylactic response within domestic rabbits (*Oryctolagus cuniculus*) and guinea pigs (*Cavia porcellus*) [151]. We particularly encourage further research with this evidence as it has a lot of promise. 

### 4.3. Modulation or Multiple Venom Systems?

Throughout the literature, it has been consistently shown that slow lorises venom is not outright lethal to arthropods or laboratory mice or conspecifics until the BGEsp and saliva are mixed together. This is still a true statement, but the saliva by itself has been toxic in its own right in some studies—in [126], the use of *Xanthonycticebus* spp. saliva on human epidermal carcinoma cells produced levels of 0.1 cell survival by measuring mitochondrial activity, demonstrating that nearly all the cancer cells were killed by the saliva. Saliva from *N. javanicus* produced inflammation in mice when they were injected with either 100 μg and 200 μg of saliva, although mice swiftly recovered and a nonsignificant amount of inflammation was observed [95]. Many slow loris bite cases of humans do not observe them licking their brachial glands, suggesting that the saliva alone could be responsible for the symptoms exhibited. Indeed, in the formative research in 1995 that first identified the need for slow loris BGE to be mixed with saliva to unlock its toxicity, some mice did die when injected with BGE only after 20 min (2/10 mice injected with extracted BGE with formic acid and 4/7 mice injected with extracted BGE with 1:1 methanol, methylene chloride (MM) [63]). Does the slow loris have the two different venoms that can be deployed depending on the function usage requirement? Or is this venom modulation? 

Cone snails (*Conus* spp.) have one of the most complex venom systems recognised by the scientific literature [2,33,152]. They are able to produce toxins more suited for predation and toxins more suited for defence at different parts along their venom duct. This morphological and physiological adaptation allows for extreme optimization of what function they require from their venom. The only other venomous species that has a similar complex system is in the assassin bugs (i.e., *Platymeris biguttatus*, *Psytalla horrida* and *Pristhesancus plagipennis*) that have anterior and posterior venom glands, with compositions varying between each gland and deployed in different scenarios (predation or defence) [153,154]. Even within species without separate glands or specific ducts, it has been shown that venomous species can have some regulation of what toxins are produced in venom even within the same gland, such as in the black-necked spitting cobra (*Naja nigricollis*) [155]. 

Some evidence for slow lorises having the ability to modulate venom comes from the brachial gland composition. More mice succumbed to BGE after 20 min when BGE was extracted with MM compared to formic acid (4/7 vs. 2/10). The justification by the author using these two different media was because formic acid could solvate hydrophilic compounds and MM could solvate lipid-based compounds [63]. The brachial gland for lorises is different to most scent glands; as identified in 2007 (discussed in Section 2.2), it has few lipids present within it and it can present as a liquid or a crystal depending on whether the individual is wild or captive ([59,76]; Nekaris, unpublished data). A lack of fats may have been intensively selected to prevent toxic activation of the BGEsp (or other unidentified compounds) before the slow loris requires it for use. If this is the case, the slow lorises may join the cone snails and assassin bugs as having the most complex venom system and venom modulation documented in the animal kingdom. 

## 5. Why Have Slow Loris Evolved Venom?

It can be seen as futile attempting to speculate on the drivers of venom within slow lorises: if we are not even certain of how, why and what venom is definitely used for, how can we discuss its evolution? Nevertheless, since the last slow loris venom review [49], new research is leading us toward some potential hypotheses of why slow lorises have evolved venom. 

### 5.1. Competition and Fitness

Many species have evolved weapons to solve disputes for mates or territories, for example, horns or antlers in Artiodactyla [156]. Slow lorises are no exception as discussed in Section 3.2, but before aggression escalates to contact, display signals, ritualised aggression and bluffing ward off actual fights. The high proportion of injuries found in wild and captive populations may indicate that lorises have frequent aggressive encounters using their venom—in *N. javanicus*, of the 76 bite sites recorded, nearly half (32 bite sites) were found on the head [59]. This causes scars and severe damage, especially to sensitive areas like the ears and eyes. This is a visual indicator of fitness—especially for more aggressive male lorises that have less fresh or healed scars, which would advertise to potential mates (and aggressors) that they are fit enough to hold onto territory and mates. Appearance is important within primates, with healthy heads being a mark of fitness, for example, within uakari monkeys (*Cacajoa* spp.) [157]. Considering venom usage as part of signalling theory, where males advertise to females, the prominence of wounds can be interpreted to be similar to honest signals, as seen in male three-spined stickleback (*Gasterosteus aculeatus*) and male mandrills (*Mandrillus sphinx*) [158,159]. If mate selection were the only reason for using venom in intraspecific encounters, we would expect it to be single sex. The pressure of both male and female dispersal, plus the resulting extreme territoriality between individuals, is what appears to drive venom usage [67]. Slow lorises demonstrate a prime area to investigate evolutionary game theory, namely the way in which weapon performance is linked to success and contest measuring [160]. 

### 5.2. Coevolution with Snakes

Snakes and primates have an entwined evolutionary history. The snake detection theory proposed by Lynne Isbell suggests that snakes are responsible for the development of primate visual systems, highlighted by the evidence in human abilities to detect snakes significantly faster amongst other “safe” images [161]. Non-human primates have also been observed in both the field and in captive settings to have strong antipredator responses to snakes—white-faced capuchin monkeys (*Cebus capucinus*) are able to detect and be vigilant against snakes from four months old [162]. Analysis of spitting cobra (*Naja* spp. and *Hemachatus hemachatus*) toxins showed a high upregulation of PLA2 toxins, cytotoxins known for enhancing pain in targets [9]. Spitting venom has evolved three times amongst the snakes, twice in species distributed across Africa and once in Asian species, with a speculative theory that this evolution coincides with the time when humans distributed across the globe. This intense arms race is cautiously suggested to be responsible for the evolution of venom within slow lorises due to behaviour, coat colouration and genetics. 

Slow lorises have a distinctive pattern on their bodies: two large teardrop-shaped patches on their eyes and a long dorsal stripe running along their spine (Figure 6). This visual display, combined with their defensive display of placing their hands above themselves and making hissing sounds, have suggested that lorises imitate cobras in their defensive display [49,60]. Further evidence is the *M1RC* gene that codes for coat colour is under purifying selection; it is being refined and mutations are suppressed [58]. Therefore, we assume that coat colouration and patterns under current selection pressures are vital to the survival of slow lorises. This theory has been discussed within the literature on multiple occasions, with it being described as Mullerian mimicry, which occurs when two animals with equivalent levels of defence (i.e., poisonous/venomous) have similar appearance or signals [49,57,59,60]. Additional evidence for this is slow lorises moving about more on the ground during the wet season when they are most at risk of death from either vehicles, humans, or predators [60]. The wet season is the period when aposematic contrast of their stripes and facemask are most prominent in *N. javanicus* [59].

Further evidence for this mimicry comes from slow lorises having independently evolved an adaptation to their α-1 nAChR receptor to be more resistant to alpha neurotoxins (particularly three fingered toxins, which are found in a large proportion of snake venom) [163]. This adaptation evolved on separate occasions within primates, most notably in strepsirrhine primates, while species within the Lorisoidea species are identified to have low binding capability with alpha-neurotoxins on their α-1 nAChR receptor, Lemuroidea have higher levels of susceptibility. Although Madagascar has venomous snakes, particularly terrestrial species belonging to Colubridae (the rear-fanged snakes) which are capable of delivering a medically significant bite, they are not regarded as life threatening [164]. The Indian and Southeast Asian distribution of slow and slender lorises, on the other hand, is conspecific to numerous life-threatening snakes. This coevolution between slow loris and Asian spitting cobras would be worth much more investigation given new evidence of the link between cobras and primates.

## 6. Conclusions

It is tempting to describe animals that develop unexpected or unexplained traits as “weird”—especially a venomous primate, a shared common ancestor, empathising our human bias when we scrutinise slow lorises. But framing slow lorises as oddities is detrimental to them, to quote conservationist Jack Ashby when discussing Australian mammals: “*We protect what we value, and if these incredible, highly adapted species are incorrectly written off as biologically inferior weirdos—cute, but ultimately evolutionarily doomed to fail—it makes conserving them much harder*” [165]. The slow lorises, as we have described here, have one of the most complex, successful, and fascinating venoms system known today.

New information has revealed that slow lorises have a specific component in the saliva (C1R) that could trigger the difficult healing we see in human bite cases, that the BGEsp not only resembles a common allergen, therefore responsible for some of the anaphylaxis we see in human bite cases, but that it also has glycosylation and binding sites, which means it could still be used for communication, and that slow lorises represent one of only a few rare examples of functional usages of venom that are rarely seen across the animal kingdom. There is still much to discover and confirm about the slow lorises and their venom system, but we have full confidence that the rapid rise of technology will enable us to ultimately discover what triggers the venom in slow lorises.

Finally, we provide here a research agenda (Appendix A) to allow expansion by future researchers, based on the information reviewed and discussed in this paper, of our knowledge on the slow loris venom system. We particularly stress the importance of long-term studies of slow lorises for behavioural and ecological evidence, which can shed light on how these venom systems work. We also encourage a completion of the identification of saliva and BGEsp from all slow loris species. These primates deserve to have their venom system understood, for the data it could offer us on venom evolution, primate evolution, immunity and potential cancer killing drugs could completely change our understanding in any of these areas.

## Figures and Tables

**Figure 1 toxins-15-00514-f001:**
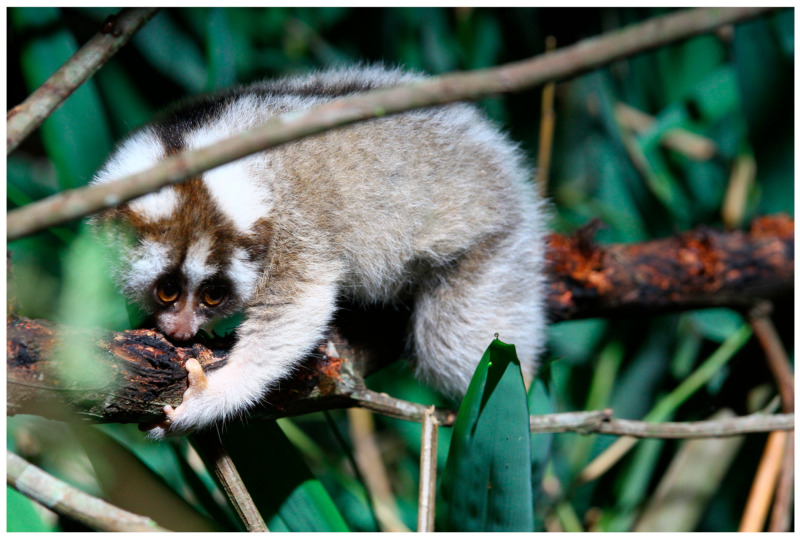
An example of a juvenile Javan slow loris (*N. javanicus*), eating exudates from a tree in Cipaganti, West Java, Indonesia, also showing gouge marks along the branch. Photo courtesy of Little Fireface Project.

**Figure 2 toxins-15-00514-f002:**
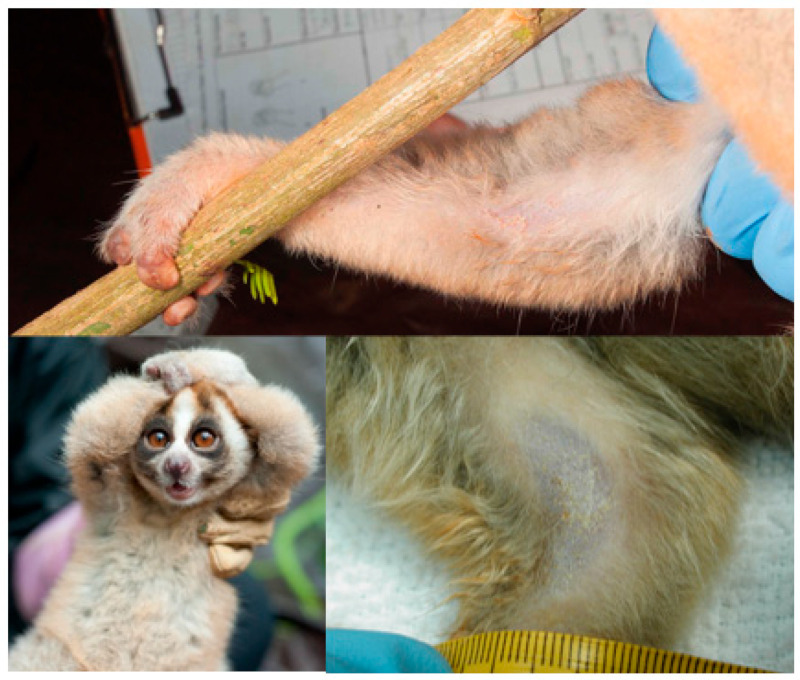
Clockwise from top: brachial gland secretions on a wild Javan slow loris (*N. javanicus*); brachial gland secretions and more bald fur on a captive pygmy loris (*Xanthonycticebus* spp.); the position of the brachial gland and the “venom pose” (*N. javanicus*). Photos of the Javan slow lorises courtesy of Little Fireface Project and captive *Xanthonycticebus* spp.photo courtesy of Christine M. Drea and Duke University Lemur Center.

**Figure 3 toxins-15-00514-f003:**
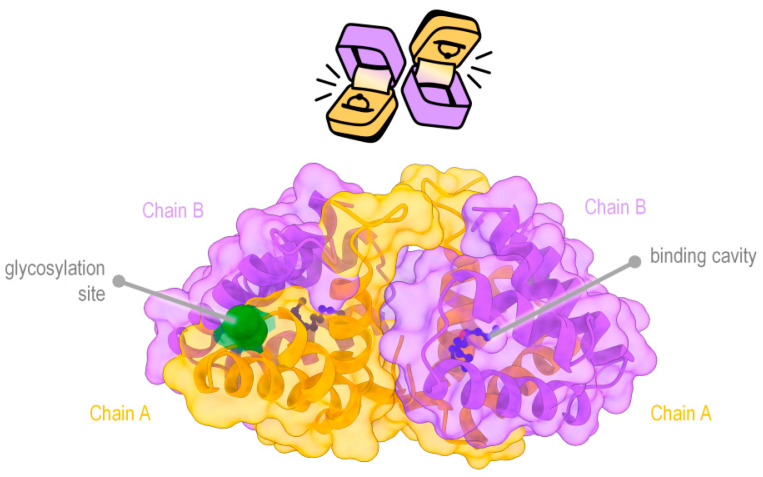
The BGE secretion protein, here Chain A is represented as orange and Chain B is represented as purple. The overall arrangement of the dimer-of-dimers is shown schematically as two jewellery boxes back to back, alluding to the ligand-binding cavity found in secretoglobins. (Protein structure was modelled using AlphaFold2 [86] and represented using UCSF ChimeraX [87] dodecyl-aldehyde is shown in the ligand pockets. Two glycosylation sites are present, with only one being visible in each “face” of the protein.)

**Figure 4 toxins-15-00514-f004:**
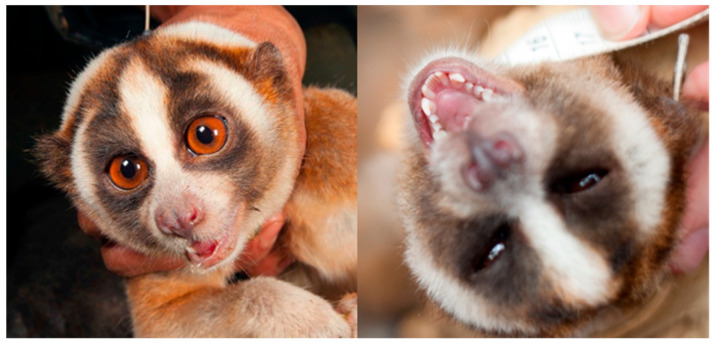
Javan slow loris (*N. javanicus*) with saliva pooling in its mouth during a health check (**left**); the slow loris toothcomb, which injects the venom (**right**). Photos courtesy of Little Fireface Project.

**Figure 5 toxins-15-00514-f005:**
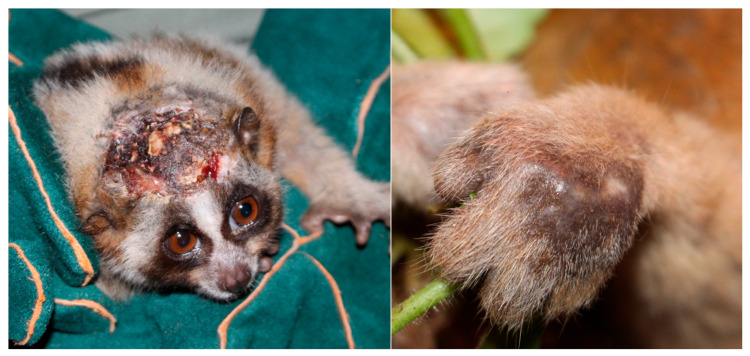
Slow-healing bites and necrotic wounds on wild Javan slow loris (*N. javanicus*), seen on the head (**left**) and on a hand (**right**).

**Figure 6 toxins-15-00514-f006:**
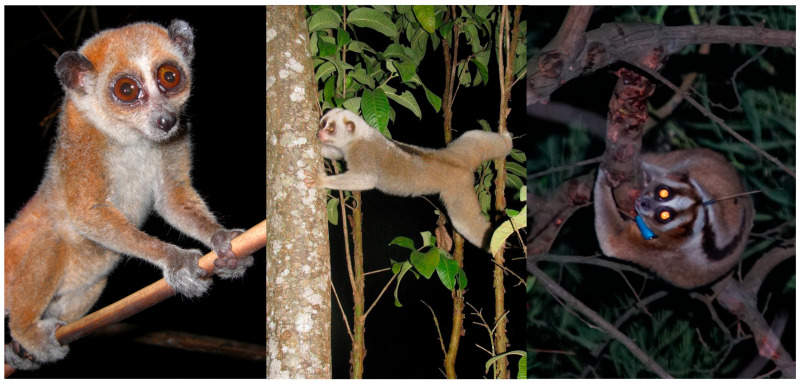
Examples of variation in the large teardrop eyes and dorsal stripe in *Xanthonycticebus pygmaeus* in Cambodia (**left**, courtesy of K.A.I. Nekaris), *N. bengalensis* in India (**middle**, courtesy of Nabajit Das) and *N. javanicus* in Java (**right**, courtesy of Little Fireface Project); the latter two individuals have also been consuming gum.

## Data Availability

BGE sequences ran through BLAST can be found in Appendix A.

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
