# Peer review of "Slowly Making Sense: A Review of the Two-Step Venom System within Slow (Nycticebus spp.) and Pygmy Lorises (Xanthonycticebus spp.)"

_toxins, 2023, doi:10.3390/toxins15090514_

Round 1
Reviewer 1 Report
The manuscript “Thinking inside the box: a venomic review of the two-step venom system within slow (Nycticebus spp.) and pygmy lorises (Xanthonycticebus spp.)” offers an exhaustive review of the venom present in slow lorises. This is a timely effort, as it’s been over 10 years since this research has been reviewed. I found the manuscript interesting and well-written, so I recommend its publication. However, I have several comments mostly related to the writing and the presentation of some ideas, that I think should be revised.
- From lines 102 through 105, the authors explain that mammals are a good model for studying venom evolution because it is an apomorphic trait, rather than plesiomorphic as observed in other animals. I strongly disagree with this statement. The only difference I see to use one or the other term is time: a plesiomorphic (ancestral) trait can also be considered apomorphic (unique to a given clade) if we focus on a deeper timescale. Look for instance at cone snails: venom in Conus geographus is plesiomorphic within conoideans, but conoidean venoms might be apomorphic within Gastropoda as most lineages are not venomous. It is usually easy to differentiate these concepts in the right context, but the way it is phrased here I think does not make all the sense.
- In line 198, the authors say the brachial gland is co-opted. Is this true? Is there evidence of this? I assumed other mammals have glands in the same place. I would be careful in separating facts from hypotheses and citing the appropriate references.
- I wonder if there is any study about the toxicity of these compounds. How much is necessary to be considered toxic and how much is produced?
- I do not think, based on how this whole section is presented, that ectoparasitic defence is a confirmed function of the venom. Of course, the authors present a lot of evidence that points in that direction, but no confirmation whatsoever.
- Lines 355-356: are mammals more sociable than, say, bees or ants? They are also venomous.
- Lines 363-365: besides observations of lorises biting and clamping their jaws in other individuals, is there evidence of they using venom? I mean, have they been observed to lick their brachial gland, for instance?
- Lines 367-371: I would argue the observation of infants dying from overgrooming is not related to the intraspecific competition. It demonstrates its effect in conspecifics, but this should be pointed out explicitly.
- Line 488: I would not go as far as to describe lorises as a “perfect” model. For starters, we don’t even know how their venom works. They are far from being a perfect model.
Thus far, I wonder if no attempts have been made to investigate the venom of lorises using transcriptomics or genomics. This is a promising avenue that could be carried out from animals in captivity and would be very informative. This line of research is presented in the “research agenda”m but that’s it. Given the prevalence of this kind of experiment in venomics, this topic should be included in the review.
Additionally, other minor comments:
- Line 22: I would avoid the use of abbreviations that have not been presented before for clarity.
- Line 97: salivary glands.
- Line 126: “slow and pygmy slow lorises”, so venomous pygmy lorises are also “slow lorises”? This is confusing, as never before (or after) the name “pygmy slow lorises” has been presented. It was “slow lorises” and “pygmy lorises”.
- Line 132: another confusing expression. “They use strength”, I’d say they use their limbs, which are strong. I think I see what the authors mean, a retractile tail is more efficient than brute force, but still.
- Line 315: resulting.
I recommend proofreading and correcting minor errors.
Author Response
Please see the table in the file - please note that in the new version we have changed the title but I cannot find a place to change this on the resubmission system (as this was important in both reviewers' comments) - the new title is
Slowly making sense: a review of the two-step venom system within slow (Nycticebus spp.) and pygmy lorises (Xanthonycticebus spp.)

Reviewer 2 Report
This manuscript reviews the loris “venom” research in an evolutionary, ecological, biological & venom components, and toxin delivery system and t is a comprehensive synthesis of published literature to date. It is a comprehensive synthesis of the slow loris. I have a few comments which should be addressed prior to acceptance of publication below. I think it is a comprehensive synthesis of the slow loris.
I am unsure as to the intent of the authors, it seems that “thinking inside the box” if they are trying to make the loris fit to a venom definition or if their intent is to say it doesn’t fit the definition. If that is the intent, challenge the definition. I would also urge you not to apply other scientists’ conjecture regarding venom systems. There are some rather broad hypothesis statements that should be reviewed and challenged, I have put some examples in my comments.
Whilst I do not agree with the current definition of venom (I think it oversimplifies a complex ecological & biological process and tries to put things in box together that aren’t really the same. Humans love to compartmentalise issues that perhaps aren’t so strictly defined). I urge caution in comparing explicit venom delivery systems over non-explicit toxin delivery systems in an evolutionary context, essentially it is comparing apples and oranges. As the authors so rightly state in the conclusion ecology and understanding the system is critical for these little studied animals and their recruitment of toxins.
What the loris has is something quite extraordinary and should be treated as such. It is only by studying this in the light of ecological and biology processes that we can understand the processes behind the recruitment of toxins and frame the findings alongside specific venom delivery systems.
Specific comments below
The title is misleading using “venomic”– you have not analysed any of the loris venom using venomics in this paper. Biological review perhaps would adequately describe the content of the manuscript.
L. 15 Grammar for clarity e.g. I think ‘compromises’ may be used incorrectly – what is being compromised? perhaps being more explicit here or using another term.
Ensure all species names are italicised in the reference list e.g. L 936, 1043, 1121
L44 the authors should also refer to the development of agricides (see Vestaron) and cosmetics (sensAmone) from venom to illustrate the breadth of bioapplications.
Please ensure you credit all species listed with their taxonomic authors on the first use of a species name and add the taxonomic reference to the references.
L65 this is a broad statement, and I would be cautious of using all lineages of species will evolve venom – just because you can do something doesn’t mean it will happen, are they evolving “venom” or are they recruiting toxins? To challenge the authors is a nudibranch can evolve it’s own venom why would they go to the bother of sequestering venom apparatus from another animal.
L77 this sentence does not make sense – humans are the “non-target” for venoms. I would suggest that for snake venom a rodent is a highly adequate model and crustacea for a marine organism is suitable but not in a rodent model. Please amend statement and clarify thoughts.
L90 I think there must be a recognition in this paragraph that these ground breaking evolution studies are predominately based in hypothesis of toxin function and not evidence-based functional characteristics of toxins and therefore these ground breaking evolution studies are built on hypothesis as opposed to evidence-based findings.
The authors need to clearly state, prior to delving into the putative function of saliva/gland components, a sentence or two for readers not familiar with lorises (almost everyone) about how they envenomate using the ‘two-step venom system’. This section needs to hone-in on the venom delivery system as per the topic of the paper. This would be ideally placed within the paragraph L126-141. This will then make the following paragraphs make sense to the reader.
The subheading 3.1 does not make sense on its own. I would recommend removing titles 3.1 and 3.2 and then just state that 3.2.1 is a hypothesised function in the first sentence of the paragraph.
L351 be wary of quoting references in a format like this as it infers those authors discovered intraspecies competition of which they did not – state clearly that the reference is a review.
L353 I am 99% sure (because biology is always unpredictable) that Cnidarian intraspecific competition has not arisen because they do not have a central nervous system as we know it. It occurs because it occurs in clonal animals in dense aggregations, fighting for room to settle in conjunction with reproduction strategies. This illustrates the importance of understanding the ecology of the animals (as you state in your conclusion) when forming evolution hypothesis. Only a handful of specimens out the 10,000 plus cnidarians are involved in intraspecies competition. This alone would infer the hypothesis to be pure conjecture. Please amend statement and remove reference. If you must use this and the above reference quite clearly state, they are reviews in the text.
Just a few minor changes of work and formatting as per in my review
Author Response

(The authors gave the same response as above.)
